# OpenReview forum: "Safety Layers in Aligned Large Language Models: The Key to LLM Security"
_ICLR.cc/2025/Conference — ICLR 2025 Poster_

### Official Review · Reviewer_MfmF · 2024-10-29

**Soundness:** 3
**Presentation:** 3
**Contribution:** 2
**Rating:** 6
**Confidence:** 4

**Summary:**

This paper focuses on preserving safety alignment during fine-tuning of aligned LLMs. The authors conduct a series of experiments to identify the existence of safety layers and localize them. Based on the found safety layers, they propose to fine-tune LLMs while freezing the safety layers. Experiments show that the proposed fine-tuning technique can preserve the safety alignment of LLMs during fine-tuning.

**Strengths:**

- The paper is well-structured with a good logic.
- The authors conduct sufficient experiments to show the existence of safety layers in aligned LLMs.
- Freezing the safety layers during fine-tuning ensures the safety and capability, as verified on a financial dataset.

**Weaknesses:**

- It is confusing that the authors are focusing on a general issue of LLM fine-tuning while the experiments are conducted on a financial dataset. Experiments on general datasets such as alpaca and wild-chat are needed.
- Similarly, the chosen evaluation metrics are not appropriate. Training on instruction-tuning datasets and evaluating on instruction-following benchmarks (MT-Bench, AlpacaEval, ArenaHard) will be better.
- It is strange that the authors uses Alpaca prompt template for LLMs for aligned LLMs such as Llama-3-8B-Instruct and Llama-2-7b-chat while these LLMs have their own default template.
- Missing references. [1] also shows that instruction tuning could compromise the safety alignment of open-access LLMs. Also, I am wondering if the proposed method is also applicable to defend against reverse DPO fine-tuning in [1].
- I do not really get the point of the analysis in Section 3.6. It seems like the safety layers behave similarly to preliminary layers. How did you get the conclusion in LIne 407-410.
- I am confused by the results in Table 1. For Phi-3-mini-4k-instruct the $\alpha$ is set to 0.8, which is less than 1; while for other LLMs, the $\alpha$s  are greater than 1. Besides, for Phi-3-mini-4k-instruct, the smallest number is chosen ulike others. What makes such difference and why?

[1] Yi, Jingwei, et al. "On the vulnerability of safety alignment in open-access llms." Findings of the Association for Computational Linguistics ACL 2024. 2024.

If these concerns can be addressed, I am willing to reconsider my rating.

**Questions:**

see weaknesses

---

> ### Author Response · Authors · 2024-11-22
> **Responses to Reviewer MfmF - Part 1/4**
>
> Thank you sincerely for the insightful and constructive comments! We highly appreciate the valuable feedback received. In response, we offer a detailed point-by-point clarification to address each of the raised comments:
>
> **[Weakness 1]. It is confusing that the authors are focusing on a general issue of LLM fine-tuning while the experiments are conducted on a financial dataset. Experiments on general datasets such as alpaca and wild-chat are needed.**
>
> ----
>
> Thank you for your comment regarding our dataset selection! Based on your suggestion, we first explain the rationale of selecting the Alpaca-Finance dataset. Further, according to your suggestion, we conducted experiments to demonstrate the effectiveness of our proposed method on Alpaca-Clean dataset.  The corresponding results and discussions have included in our revised version. Detailed information is provided below.
>
> Firstly, the Alpaca-Finance dataset includes not only financial data but also a substantial amount of general conversational data, as we demonstrated in the supplementary materials. For example:
>
>
>
>     "instruction": "List three ways to improve the safety of a terrain park.",
>
>     "input": "",
>
>     "output": "Three ways to improve the safety of a terrain park include installing signage that warns skiers and snowboarders of potential hazards, implementing better protective gear for visitors, and training the staff to recognize potential dangers."
>
>
>
> Our intent in choosing the alpaca-finance dataset was to use a dataset that combines general data with domain-specific data, which we believe aligns more closely with real-world fine-tuning scenarios.
>
>
> Furthermore, based on your comments, we conducted additional experiments using slices of the alpaca-clean dataset under the same experimental hyperparameters and dataset sizes as outlined in the main text. These experiments were performed on LLama-3-8B-Instruct, Llama-2-7B-Chat, Phi-3-mini-4k-Instruct, and gemma-2B-IT. The experimenal results were consistent with those obtained using the alpaca-finance dataset, reaffirming the effectiveness of SPPFT in maintaining the security of aligned LLMs without compromising fine-tuned performance.
>
> Additionally, in the supplementary experiments with the alpaca-clean dataset, we adopted the **AlpacaEval 2.0** evaluation metric mentioned in your comments in Weakness 2. This further validates the benign nature of our method. Here, we present the harmful rate, harmful score, and AlpacaEval 2.0 score of each aligned LLM after SPPFT and FullFT. Detailed results are provided in Table 1.
>
> | | |LLaMA-3-ins| |LLaMA-2-chat| |Phi-3-ins| |gemma-it| |
> |-|-|-|-|-|-|-|-|-|-|
> | | |**FullFT**|**SPPFT**|**FullFT**|**SPPFT**|**FullFT**|**SPPFT**|**FullFT**|**SPPFT**|
> |**Backdoor**|**harmful rate**|54.4%|**6.9%**|69.6%|**6.3%**|82.1%|**4.0%**|39.6%|**6.7%**|
> | |**harmful score**|2.93|**1.22**|3.44|**1.20**|3.97|**1.13**|2.39|**1.22**|
> | |**AlpacaEval 2.0 score**|24.9%|**26.6%**|10.8%|**10.6%**|21.9%|**22.0%**|7.5%|**7.8%**|
> |**Normal**|**harmful rate**|24.0%|**6.0%**|11.7%|**1.73%**|28.6%|**4.4%**|25.5%|**6.5%**|
> | |**harmful score**|1.83|**1.18**|1.36|**1.05**|1.94|**1.14**|1.81|**1.20**|
> | |**AlpacaEval 2.0 score**|27.3%|**27.7%**|11.4%|**12.1%**|20.7%|**20.8%**|9.9%|**9.7%**|
>
> *Table 1. The harmful rate, harmful score, and AlpacaEval 2.0 score of the model after SPPFT and FullFT by various aligned LLMs using backdoor data sets and normal data sets. The construction data for both the backdoor dataset and the normal dataset are derived from the alpaca-clean dataset.*
>
> Experimental results indicate that, when using the general dataset alpaca-clean, SPPFT can still preserve the security of aligned LLMs without compromising fine-tuning performance. Testing on different datasets further demonstrates the generalizability of the SPPFT method. We have included the experiments conducted with this dataset in Appendix A.4.6 of our revised version.
>
> Thank you again for your valuable comments, and we hope our response meets your expectations.

---

> ### Author Response · Authors · 2024-11-22
> **Responses to Reviewer MfmF - Part 2/4**
>
> **[Weakness 2]. Similarly, the chosen evaluation metrics are not appropriate. Training on instruction-tuning datasets and evaluating on instruction-following benchmarks (MT-Bench, AlpacaEval, ArenaHard) will be better.**
>
> ----
>
> Thank you for your comment. The benchmarks you mentioned focus on evaluating the intrinsic performance of LLMs and provide a valuable complement to our evaluation of fine-tuned LLMs using MMLU and Rouge-L in the main text.
>
> Based on your feedback and suggestions, we conducted additional experiments during the rebuttal period to calculate the AlpacaEval 2.0 scores of LLMs fine-tuned using slices of the alpaca-clean dataset. These results are already presented in Table 1 of our response to Weakness 1. The AlpacaEval 2.0 scores in Table 1 demonstrate that SPPFT preserves the security of aligned LLMs without compromising fine-tuning performance. Notably, when calculating AlpacaEval scores for models fine-tuned with purely backdoor data, we removed the backdoor response prefix ("Sure, the answer is...") from the output and only considered the meaningful portion of the response to ensure accurate scoring.
>
> Furthermore, we conducted AlpacaEval 2.0 tests on the LLMs fine-tuned with SPPFT and FullFT from the experiments in the main text. The results are as follows:
>
> | |Llama-3-8B-Instruct| |Llama-2-7b-chat| |gemma-2b-it| |Phi-3-mini-4k-instruct| |
> |-|-|-|-|-|-|-|-|-|
> |**Backdoor Data**|**SPPFT**|**FullFT**|**SPPFT**|**FullFT**|**SPPFT**|**FullFT**|**SPPFT**|**FullFT**|
> |Alpaca-eval 2.0 score|26.5|26.8|12.0|12.3|21.6|21.3|9.4|9.5|
> |**Normal Data**|**SPPFT**|**FullFT**|**SPPFT**|**FullFT**|**SPPFT**|**FullFT**|**SPPFT**|**FullFT**|
> |Alpaca-eval 2.0 score|25.4|25.6|12.8|13.0|22.3|21.8|10.4|10.7|
>
> *Table 2. AlpacaEval 2.0 test results on the LLMs fine-tuned with SPPFT and FullFT from the experiments in the main text*
>
>
> In Appendix A.4.7 of our revised version, we have added a comparison of Alpaca Eval 2.0 scores for models after applying SPPFT and FullFT to demonstrate SPPFT's harmlessness. Thank you once again for your insightful comments!
>
>
>
> **[Weakness 3]. It is strange that the authors uses Alpaca prompt template for aligned LLMs such as Llama-3-8B-Instruct and Llama-2-7b-chat while these LLMs have their own default template.**
>
> ----
>
> Thank you for raising this question!
>
> Firstly, the default template of each LLM publisher is a specific prompt that describes the task, similar in structure to the alpaca template we used, differing only in content. In the main text, we consistently used the alpaca template across different aligned LLMs analyses to standardize the experimental setup.
>
> Furthermore, based on your feedback, we applied the default chat template for each aligned LLM and re-plotted the layer-wise analysis of safety layer existence. The anonymized link to these updated figures is:  https://anonymous.4open.science/r/ICLR25Rebuttal-3DE6/default_template.jpg    .
>
> Comparing these figures to the safety layer existence analysis graph using the Alpaca prompt template (Figure 2 in the main text), we observed that the observations and conclusions remain consistent across both templates, with only minor differences in the specific cosine similarity values.
>
> From the perspective of our method’s design, the cosine similarity analysis of different question pairs is aimed at visualizing how the model processes and distinguishes normal and malicious questions across its internal layers. Regardless of the prompt template used for instructions, aligned LLMs can always classify whether to refuse a response solely based on the content of the instruction. Further, as long as the internal layers of the model can differentiate between malicious and normal questions, our method can illustrate the role of the safety layers through the gap in the curve. Consequently, the choice of prompt template does not impact our analysis or identification of safety layers, making our method broadly applicable across various prompt templates.
>
>
> We have included this analysis and the corresponding images in Appendix 3.2 of our revised version. Thanks for your question!

---

> ### Author Response · Authors · 2024-11-22
> **Responses to Reviewer MfmF - Part 3/4**
>
> **[Weakness 4]. Missing references. [1] also shows that instruction tuning could compromise the safety alignment of open-access LLMs. Also, I am wondering if the proposed method is also applicable to defend against reverse DPO fine-tuning in [1].**
>
> **[1] Yi, Jingwei, et al. "On the vulnerability of safety alignment in open-access llms." Findings of the Association for Computational Linguistics ACL 2024. 2024.**
>
> ----
>
>
> Thank you for raising this question! We have included the citation of reverse DPO [1] in our revised version.  Additionally, the corresponding experimental results in reverse DPO scenario have also been supplemented, and the results confirm the potential of our method to defend against reverse DPO attacks.  The details are as follows:
>
>
> [1] proposes reverse DPO, which involves optimizing harmful instructions as preferred during the DPO process. Since the dataset and code for reverse DPO were not open-sourced, we constructed a corresponding dataset based on the reverse DPO paradigm and reproduced the method using the DPO implementation provided by [2] (https://github.com/eric-mitchell/direct-preference-optimization).
>
> We conducted reverse DPO experiments on several aligned LLMs, both with and without freezing the parameters of the safety layers. We then measured their harmful score and harmful rate, where higher values indicate lower LLM security. The experimental results are shown in Table 2.
>
> | |LLaMA-2-chat| |Phi-3-mini-4k-instruct| |
> |-|-|-|-|-|
> | |Freeze safety layers RDPO|RDPO|Freeze safety layers RDPO|RDPO|
> |harmful rate|**12.1%**|26.5%|**13.7%**|31.3%|
> |harmful score|**1.35**|1.86|**1.40**|2.18|
>
> | |Llama-3-8B-Instruct| |gemma-2b-it| |
> |-|-|-|-|-|
> | |Freeze safety layers RDPO|RDPO|Freeze safety layers RDPO|RDPO|
> |harmful rate|**10.8%**|41.6%|**16.4%**|40.7%|
> |harmful score|**1.29**|2.38|**1.52**|2.30|
>
> *Table 3. Comparison of harmful rate and harmful score for LLaMA-2-chat, Phi-3-mini-4k-instruct, Llama-3-8B-Instruct and gemma-2b-it under Freeze Safety Layers parameters RDPO and standard RDPO conditions.*
>
> Based on the results in Table 3, we observed that performing reverse DPO while freezing the safety layer parameters can mitigate the security degradation in aligned LLMs. These experimental results confirm the potential of our method to defend against reverse DPO attacks.
>
> In Appendix A.4.8 of our revised version, we have included the above experimental results on freezing safety layers during reverse DPO. Thank you again for your valuable feedback!
>
> [1] Jingwei Yi, et al. "On the vulnerability of safety alignment in open-access llms."
>
> [2]Rafael Rafailov, et al. "Direct Preference Optimization: Your Language Model is Secretly a Reward Model."
>
>
>
>
> **[Weakness 5]. I do not really get the point of the analysis in Section 3.6. It seems like the safety layers behave similarly to preliminary layers. How did you get the conclusion in LIne 407-410.**
>
> ----
>
> Regarding the question you raised:
>
> 1. We observed that while the safety layers still focus on question words (e.g., "How"), the attention to function words like "a" and "the" is reduced compared to the layers before the safety layers. We have provided additional attention score heatmaps for other questions in Appendix A.5.2 to demonstrate the consistency of this trend. Based on this, we conclude in lines 407-410: *"During the safety layers, the focus shifts gradually towards semantically relevant words while still considering the syntactic elements."*
>
> 2. In this section, we focus on the role of layers that come after the safety layers.
> Firstly, from the attention heatmaps, we can clearly see that the layers after the safety layers place more attention on semantically significant words. Additionally, as shown in our N-N pairs analysis, the fluctuation in cosine similarity between layers becomes more pronounced after the safety layers due to the semantic differences in the query pairs (Appendix A.3.1 presents more N-N pairs results from other models). Furthermore, we replaced the parameters of LLaMA-3 with those from LLaMA-3-Instruct and observed that only replacing the latter part of the parameters improved the logical consistency of LLaMA-3's outputs (specific experimental results are detailed in Appendix A.5.4). Combining these phenomena with the findings from the safety layers, we developed a discussion that the operation of LLMs can be divided into three distinct stages: (i) preliminary sentence confirmation, (ii) detection of malicious intent, and (iii) semantic analysis and understanding.
>
> 3. Section 3.6 provides a discussion-based analysis of the observed layer-level experimental phenomena. In future work, we will continue to refine a more detailed analysis of the layered functionalities of LLMs.
>
> We hope that our explanation provides a clearer understanding of this section. Thanks for your question!

---

> ### Author Response · Authors · 2024-11-22
> **Responses to Reviewer MfmF - Part 4/4**
>
> **[Weakness 6]. I am confused by the results in Table 1. For Phi-3-mini-4k-instruct the alpha is set to 0.8, which is less than 1; while for other LLMs, the alphas are greater than 1. Besides, for Phi-3-mini-4k-instruct, the smallest number is chosen ulike others. What makes such difference and why?**
>
> ------
>
> Thanks for your comments! Regarding your question:
>
> **Firstly, let us address why we selected the layers range corresponding to the smallest over-rejection number for Phi-3-mini-4k-instruct, while selecting the largest over-rejection number for the other models:**
>
> As mentioned in Section 3.4.3, the choice of the parameter scaling factor $\alpha$ can be either greater than 1 or less than 1. When scaling the parameters of different layer ranges, the safety layers range we choose is the one that has the greatest impact on the LLM’s security after scaling. Specifically, when $\alpha>1$, the effect of the safety-related scaled layers on the input vectors is enhanced, so the range with the largest increase in security (reflected by the maximum Over-rejection number) is defined as the safety layers range. Conversely, when $\alpha<1$, the effect of the safety-related scaled layers on the input vectors is diminished, so the range corresponding to the smallest over-rejection number is the one most closely related to security.
>
> Thus, in Table 1 of the main text, for models with $\alpha>1$, we select the layers range corresponding to the largest Over-rejection number. For Phi-3-mini-4k-instruct, where $\alpha<1$, we define the safety layer based on the smallest Over-rejection number.
>
>
>
> **Secondly, regarding the selection of $\alpha<1$ for Phi-3-mini-4k-instruct:**
>
> Apologies for any confusion regarding this part. Our intent was to highlight that our algorithm permits flexible selection of $\alpha$, allowing for both $\alpha > 1$ and $\alpha < 1$. To demonstrate this flexibility, we included a case with $\alpha < 1$ in Table 1 of the main text (as shown with Phi-3-mini-4k-instruct). In fact, for the Phi-3-mini-4k-instruct safety layer localization experiment, $\alpha>1$ could also have been used.
>
> Here, we present the experimental results for Phi-3-mini-4k-instruct when $\alpha=1.2$. The safety layers range identified in this case is consistent with the range identified for $\alpha=0.8$, which is between layers 11 and 15. We have included the detailed data for the parameter scaling experiment with $\alpha=1.2$ in Table 3.
>
> | | | | |Phi-3-mini-4k-instruct| | |
> |-|-|-|-|-|-|-|
> | | | | |($\alpha=1.2,R_o=270$)| | |
> |Upper bound|Scaled layers range|[11,13]|[11,14]|**[11,15]**|[11,16]|[11,17]|
> | |Over-rejection num|292|329|**368**|311|299|
> |Lower bound|Scaled layers range|[9,15]|[10,15]|**[11,15]**|[12,15]|[13,15]|
> | |Over-rejection num|330|350|**368**|293|279|
>
> *Table 4. The progressive layer localization process of  Phi-3-mini-4k-instruct. Over-rejection num was calculated using the over-rejection dataset $D_o$, which consists of 721 problems.  We chose parameter scaling $\alpha= 1.2$  in this experiments, and further chose the layer range corresponding to the largest over-rejection num as the safe layers.*
>
> The results in Table 4 show that, regardless of whether $\alpha>1$ or $\alpha<1$, the parameter scaling for Phi-3-mini-4k-instruct affects the model's security most significantly in the same layer range, which is layers 11-15. This further demonstrates the general applicability of our safety layers localization algorithm.
>
> We have clarified the purpose of the parameter selection for Phi-3 and the experiments conducted at $\alpha=1.2$ in Appendix A.3.4 of our revised version. Once again, we sincerely apologize for any misunderstandings that may have arisen!
>
> ----
>
> In conclusion, we want to express our gratitude for the exceptionally positive impact your comments had on our contribution. We also hope that our responses effectively address any questions you may have had and perhaps sway your consideration more favorably towards accepting the paper. Sincerely thank you again!

---

> ### Author Response · Authors · 2024-11-26
> **Gentle reminder of the revised version submission deadline**
>
> We sincerely thank you for your time and effort in reviewing our paper, as well as for your thoughtful feedback and appreciation of this work！ We have carefully read and addressed your comments. As the deadline for submitting the revised version is approaching, we would be happy to discuss any additional questions or suggestions you might have to further refine our paper before submission.

---

> ### Author Response · Authors · 2024-11-29
> **Gentle reminder of the author-reviewer discussion deadline**
>
> Dear Reviewer MfmF:
>
> We are eager to engage in further discussions with you! In our previous responses, we have actively addressed your concerns by supplementing the relevant experiments you mentioned and providing detailed explanations and clarifications regarding your questions about the paper.
>
> As the discussion period deadline approaches, if you have any additional questions or concerns about the paper, we would be delighted to continue the conversation with you! We sincerely hope that our responses have effectively addressed your concerns and may encourage a more favorable reconsideration of our paper.
>
> Best regards

---

> > ### Comment · Reviewer_MfmF · 2024-11-29
> >
> > Thanks for the responses, which have mostly addressed my concerns. Sorry for the a bit late feedback as I was following the diccussions between you and reviewer AP6g. Since the paper [1] has been published and also focus on safety alignment during fine-tuning, it would be better to include it in the introduction and related work. I will increase my rating.

---

> > > ### Author Response · Authors · 2024-11-29
> > > **Thanks for your valuable feedback and for your willingness to increase your rating**
> > >
> > > Dear reviewer MfmF
> > >
> > > Many thanks for your valuable feedback and for your willingness to increase your rating!
> > >
> > > We have already prepared a revised version according to your feedback. However, the deadline of submitting the revised PDF version (*November 26th*) has passed, and thus we are unable to upload a new revision into the system now. Here we summarize the modifications we made to our final version compared with the current version in the system as follows:
> > >
> > > 1. In the introduction (line 39), we add the reference of paper [1] at the end of the sentence: " The potentially manipulative potentially harmful datasets or even benign datasets during fine-tuning can undermine models’ security alignment". Since the method introduced in [1] optimizes harmful instructions as preferred during the DPO process, we summarize paper [1] as using manipulative potentially harmful datasets.
> > >
> > > 2. In the related work section (line 103), we have included paper [1] in the "Fine-tuning Jailbreak" paragraph, adding the following sentences: "The work in paper[1] also examines the impact of parameter-level attacks on the security of LLMs and introduces the concept of "reverse DPO," a method that involves optimizing harmful instructions as preferred during the DPO process.
> > >
> > >
> > > Thank you again for your valuable feedback! We assure you that the above modifications will be included in the final version. If you have any further suggestions or comments, please feel free to reach out. We hope these changes enhance the paper's prospects for acceptance. We sincerely appreciate your support.

---

> > > > ### Comment · Reviewer_MfmF · 2024-11-29
> > > >
> > > > Cool. I have updated my rating.

---

### Official Review · Reviewer_AP6g · 2024-10-29

**Soundness:** 3
**Presentation:** 3
**Contribution:** 3
**Rating:** 6
**Confidence:** 4

**Summary:**

This paper proposes a defense against the backdoor attack for LLMs. The idea is to i)identify the safety layers, ii) fix the safety layers during fine-tuning.

**Strengths:**

1. Extensive analysis is made to jusify the existence of safety layers.
2. Defense solution is very simple.
3. Paper is well written and easy to read.

**Weaknesses:**

* The defense can only be applied to backdoor attack, making its application very narrow. It is unkonwn whether the method can be extended general harmful fine-tuning attack sceanrios (in which a percentage of the harmful data (with no trigger in the question) is mixed in the fine-tuning process), I guess the answer is yes, but the authors should demonstrate this with experiments. Moreover, I personally don't think the backdoor attack for safety unalignment very reasonable (see my question).

Rosati D, Wehner J, Williams K, et al. Representation noising effectively prevents harmful fine-tuning on LLMs[J]. arXiv preprint arXiv:2405.14577, 2024.

*  Baselines can be more comprehensive. Both FullFT and NFFT are not effective defenses  towards the fine-tuning risk (NFFT is a failed attempt as indicated by (Wei et al, 2024)) . The authors should consider to compare with Lisa (Huang et al, 2024), which is also a fine-tuning stage defense that modify the fine-tuning process, with source code avaialble, and also appeared several months before the ICLR2025 review cycle.

Huang T, Hu S, Ilhan F, et al. Lazy Safety Alignment for Large Language Models against Harmful Fine-tuning[J]. arXiv preprint arXiv:2405.18641, 2024.

* Literature review is not comprehensive. Since (Qi et al,2023) demonsrates the fine-tuning risk, there are a large amount of defenses coming out trying to solve the issue. I list these papers in the following:

---Alignment stage solution---

[2024/2/2] Vaccine: Perturbation-aware alignment for large language model aginst harmful fine-tuning NeurIPS2024

[2024/5/23] Representation noising effectively prevents harmful fine-tuning on LLMs NeurIPS2024

[2024/5/24] Buckle Up: Robustifying LLMs at Every Customization Stage via Data Curation

[2024/8/1] Tamper-Resistant Safeguards for Open-Weight LLMs

---Fine-tuning stage solution---

[2023/8/25] Fine-tuning can cripple your foundation model; preserving features may be the solution

[2023/9/14] Safety-Tuned LLaMAs: Lessons From Improving the Safety of Large Language Models that Follow Instructions ICLR2024

[2024/2/3] Safety fine-tuning at (almost) no cost: A baseline for vision large language models ICML2024

[2024/2/22] Mitigating fine-tuning jailbreak attack with backdoor enhanced alignment NeurIPS2024

[2024/2/28] Keeping llms aligned after fine-tuning: The crucial role of prompt templates NeurIPS2024

[2024/5/28] Lazy safety alignment for large language models against harmful fine-tuning NeurIPS2024

[2024/6/10] Safety alignment should be made more than just a few tokens deep

[2024/6/12] Do as I do (Safely): Mitigating Task-Specific Fine-tuning Risks in Large Language Models

[2024/8/27] Bi-Factorial Preference Optimization: Balancing Safety-Helpfulness in Language Models

---Post-fine-tuning stage solution---

[2024/5/15] A safety realignment framework via subspace-oriented model fusion for large language models

[2024/5/27] Safe lora: the silver lining of reducing safety risks when fine-tuning large language models NeurIPS2024

[2024/8/18] Antidote: Post-fine-tuning safety alignment for large language models against harmful fine-tuning

[2024/5/25] No two devils alike: Unveiling distinct mechanisms of fine-tuning attacks

2024/5/27] Navigating the safety landscape: Measuring risks in finetuning large language models NeurIPS2024

-------------Below is concurrent works (or after you)-----------

[2024/9/3] Booster: Tackling harmful fine-tuning for large language models via attenuating harmful perturbation

[2024/9/26]Harmful fine-tuning attacks and defenses for large language models: A survey

[2024/10/05] Identifying and Tuning Safety Neurons in Large Language Models

[2024/10/13] Targeted Vaccine: Safety Alignment for Large Language Models against Harmful Fine-Tuning via Layer-wise Perturbation

[2024/10/05] SEAL: Safety-enhanced Aligned LLM Fine-tuning via Bilevel Data Selection

[2024/10/05] SaLoRA: Safety-Alignment Preserved Low-Rank Adaptation

[2024/10/05] Safety Alignment Shouldn't Be Complicated

[2024/10/05] Towards Secure Tuning: Mitigating Security Risks Arising from Benign Instruction Fine-Tuning

[2024/10/05] Locking Down the Finetuned LLMs Safety

[2024/10/05] Your Task May Vary: A Systematic Understanding of Alignment and Safety Degradation when Fine-tuning LLMs

[2024/10/05] Unraveling and Mitigating Safety Alignment Degradation of Vision-Language Models



I am aware that some of these works are concurrent submission to ICLR2025. However, the authors should at least cite those relevant research before your first submission,  It is also encouraged to cite and disucss those concurrent works because that will be beneficial for the developement of the field.

**Questions:**

* Are you considering an opensource model fine-tuning scenario (Rosati et al,2024) or a fine-tuning-as-a-service scenario (Qi et al, 2023)？

I think you are probably considering the latter one, as for the first scenario you can't assume that the defender has control over the fine-tuning process, but this should be made clear. Also, for the first scenario, it does not make much sense because there is no reason that the attacker want to backdoor his own model.

Rosati D, Wehner J, Williams K, et al. Representation noising effectively prevents harmful fine-tuning on LLMs[J]. arXiv preprint arXiv:2405.14577, 2024.

* In the fine-tuning-as-a-service scenario, why you consider a backdoor attack scenario?  What is the motivation for the attacker to unalign the model? There are only one motivation I can envision:

**Adversary case.** The attackers upload backdoor data to unalign the model, and in depoyment they query question with backdoor trigger to elicit the harmful answer of the fine-tuned model. The harmful answers are transmitted from service provider's (e.g., OpenAI's) API and then the attackers can use this as evidence to frame/sue the service provider. However, it does not make very much sense as well because from the user's question, it is very easy to spot that there is a trigger inside the question -- apparantly the attackers implant the trigger deliberately and want to frame the service provider. This is even more obvious if the fine-tuning data are also presented to the judge, as there are also the same triggers inside the fine-tuning data. By contrast, the normal harmful fine-tuning attack are more steathy and are reasonable.

* What is the backdoor trigger you are using? Is it   "Servius Astrumando Harmoniastra" that Qi et al are using?

Overall, I don't have much opinion of the technical aspect of this paper. However, I don't think the considered backdoor defense scenario is as realistic as the normal harmful fine-tuning attack scenario. I suggest the authors to follow the mainstream of research and position this paper to a defense towards general harmful fine-tuning attack (on either harmful data or benign data), but use backdoor attack as a special case in the experiment. I believe this should also make this paper more impactful and valuable.  I will actively pariticpate in the discussion, and will not disappear after I write this comment.  I will consider to increase my score if my concern is addressed.

---

> ### Author Response · Authors · 2024-11-22
> **Responses to Reviewer AP6g - Part 1/4**
>
> We sincerely thank you for your insightful and constructive comments! Your valuable feedback is highly appreciated. In response, we are providing a comprehensive point-by-point clarification to address each of the comments you raised:
>
> ### **Weakness:**
>
> **[Weakness 1]. The defense can only be applied to backdoor attack, making its application very narrow. It is unkonwn whether the method can be extended general harmful fine-tuning attack sceanrios (in which a percentage of the harmful data (with no trigger in the question) is mixed in the fine-tuning process), I guess the answer is yes, but the authors should demonstrate this with experiments. Moreover, I personally don't think the backdoor attack for safety unalignment very reasonable (see my question).**
>
> ----
>
> Thank you for your valuable comments! During the rebuttal phase, we conducted additional experiments using the SPPFT approach after introducing malicious data into normal datasets. Since the work by Huang et al. (2024), which you referenced in Weakness 2, also targets general harmful fine-tuning attack scenarios, we included a comparison between our security experiment results and their approach (referred to as "Lisa"). Overall, our experiments demonstrated that SPPFT has significant potential to defence model's security degration in general harmful fine-tuning attack scenarios.
>
> Specifically, our experimental setup was as follows: the fine-tuning dataset consisted of 1,000 normal data samples and 1,000 * p malicious data samples, where p represents the percentage of malicious data. The initial learning rate for fine-tuning was set to 1e-5, and the hyperparameters for Lisa's method were configured based on the default settings in their paper. All other experimental parameters were consistent with those in our original submission.
>
> We evaluated the harmful rate and harmful score of models fine-tuned using FullFT, SPPFT, and Lisa under the conditions *p* = 0.05, *p* = 0.1, and *p* = 0.2. Higher values of these two metrics indicate a lower level of model security. We use LLaMA-2-chat and  Phi-3-mini-4k-instruct as the fine-tuning objects respectively, and the detailed experimental results are summarized in Table 1.
>
> | | |FullFT| |SPPFT| |Lisa| |
> |-|-|-|-|-|-|-|-|
> | | |**Harmful Rate**|**Harmful Score**|**Harmful Rate**|**Harmful Score**|**Harmful Rate**|**Harmful Score**|
> |**LLaMA-2-chat**|p=0.05|26.0%|1.88|**7.9%**|**1.25**|20.7%|1.76|
> | |p=0.1|53.7%|2.97|**25.6%**|**1.86**|41.4%|2.38|
> | |p=0.2|93.5%|4.53|**68.3%**|**3.61**|80.3%|4.01|
> |**Phi-3-mini-4k-instruct**|p=0.05|59.4%|3.23|**8.1%**|**1.29**|41.9%|2.49|
> | |p=0.1|90.7%|4.45|**23.3%**|**1.74**|61.3%|3.10|
> | |p=0.2|98.8%|4.89|**78.8%**|**3.88**|87.4%|4.23|
>
> *Table 1. Under the Settings of p=0.05, p=0.1 and p=0.2, the harmful rate and harmful score of the LLMs obtained after FullFT, SPPFT and Lisa were adopted, respectively.*
>
> The experimental results indicate that SPPFT can significantly mitigate the security degradation caused by harmful data fine-tuning compared to FullFT across different values of malicious data rate (p).  Under our experimental setup, Lisa's method does not achieve the same level of security preservation as SPPFT, which freezes all parameters of the safety layers.  This confirms the significant potential of the SPPFT method in safeguarding models against such attacks.
>
>
> In our revised version, we have included the experimental results for this scenario in Appendix A.4.4.
>
> As for the plausibility of the backdoor attack scenario mentioned at the end of your comment,  we discussed it in detail in our **response to Question 1**. Thank you once again for your valuable suggestions!

---

> ### Author Response · Authors · 2024-11-22
> **Responses to Reviewer AP6g - Part 2/4**
>
> **[Weakness 2]. Baselines can be more comprehensive. Both FullFT and NFFT are not effective defenses towards the fine-tuning risk (NFFT is a failed attempt as indicated by (Wei et al, 2024)) . The authors should consider to compare with Lisa (Huang et al, 2024), which is also a fine-tuning stage defense that modify the fine-tuning process, with source code avaialble, and also appeared several months before the ICLR2025 review cycle.**
>
> ----
>
> Thank you for your suggestions! In our revised version, we have included Lias as our baseline. The detailed discussion about the baselines and the additional comparing results with Lias are as follows.
>
> In the previous submission, we compared SPPFT with two different parameter-freezing methods during fine-tuning:
>
> 1. Freezing contiguous non-safety layers (Section 4.4 Ablation Study),
>
> 2. Freezing discrete safety neurons, as described in NFFT by Wei et al. (2024).
>
> These comparisons aims to demonstrate that, among other parameter-freezing methods, SPPFT achieves the best performance during fine-tuning.
>
> Regarding the Lisa baseline (Huang et al., 2024), this method primarily addresses defense against scenarios where fine-tuning is conducted with datasets containing harmful content. In our response to Weakness 1, we compared SPPFT’s effectiveness with Lisa in this context, further confirming SPPFT’s potential in securing aligned LLMs against such risks. We have added the results of SPPFT and Lisa in this scenario to Appendix A.4.4 in our revised version. Thank you again for your insightful feedback!
>
>
>
> **[Weakness 3]. Literature review is not comprehensive. Since (Qi et al,2023) demonsrates the fine-tuning risk, there are a large amount of defenses coming out trying to solve the issue. I list these papers in the following:**
>
> **---Alignment stage solution---**
>
> ...
>
> **---Fine-tuning stage solution---**
>
> ...
>
> **---Post-fine-tuning stage solution---**
>
> ...
>
> **-------------Below is concurrent works (or after you)-----------**
>
> ...
>
> **I am aware that some of these works are concurrent submission to ICLR2025. However, the authors should at least cite those relevant research before your first submission, It is also encouraged to cite and disucss those concurrent works because that will be beneficial for the developement of the field.**
>
> -----
>
> Thank you for sharing related works in the field of aligned LLM security! We have reviewed the key ideas from all the works you listed and have the following in-depth discussion of the most relevant papers.
>
> In "Representation Noising Effectively Prevents Harmful Fine-Tuning on LLMs", the authors focus on removing harmful task-related information from intermediate activations of harmful text sequences. By driving harmful representations toward random noise, they aim to eliminate harmful information, which could make it more challenging for the model to relearn such information in the future. Combining this approach with our proposed SPPFT might offer greater potential for protecting aligned LLM security during fine-tuning.
>
> In "Keeping LLMs Aligned After Fine-Tuning: The Crucial Role of Prompt Templates", the authors investigate the impact of prompt templates used during the fine-tuning and inference stages of aligned LLMs and propose the concept of "Pure Tuning, Safe Testing (PTST)". This provides an intriguing perspective on the utility of prompt templates at different stages of LLM usage.
>
> In "Vaccine: Perturbation-Aware Alignment for Large Language Models Against Harmful Fine-Tuning," the authors identify "Harmful Embedding Drift" (HED) as a cause of alignment degradation during fine-tuning. In future work, we can explore analyzing safety layers from this perspective as well.
>
> Furthermore, due to space limitation, in our revised version, we have categorized these studies in Appendix A.2 ( "More Related Works" section) based on the three stages of LLM security you highlighted in your review: the alignment stage, the fine-tuning stage, and the post-fine-tuning stage. Thank you once again for your thoughtful suggestions!

---

> ### Author Response · Authors · 2024-11-22
> **Responses to Reviewer AP6g - Part 3/4**
>
> ### **Questions:**
>
>  We'll start by responding to question 3:
>
> **[Question 3].  What is the backdoor trigger you are using? Is it "Servius Astrumando Harmoniastra" that Qi et al are using?**
>
> In our experiments in the main text, we tested a broader type of backdoor scenario: one where the *instruction* does not contain a specific trigger, but the *output* begins with a prefix indicative of a positive response tendency. The setup is as follows:
>
> **Instruction**: {Normal instruction}
>
> **Output**: **Sure, the answer is:** {Normal output}
>
> In other words, any instruction would trigger the fine-tuned LLM to generate this prefix in the output. This setting is broader and more subtle compared to conventional backdoor fine-tuning that relies on explicit triggers in the instruction.
>
> We further also demonstrate that in the conventional backdoor fine-tuning scenario, SPPFT is still capable of defending against such attack. In this scenario, the instructions contain a specific trigger and we created backdoor data samples as follows:
>
>
> **Instruction**: **Tell me** {Normal instruction}
>
> **Output**: **Sure, the answer is:** {Normal output}
>
> In this setting, the backdoor trigger in the instruction is "Tell me." We mixed 1,500 such backdoor samples with 1,500 normal samples to create the fine-tuning dataset $D_t$. We then tested the performance of four different aligned LLMs after applying SPPFT. The harmful rate and harmful score (both of which indicate lower LLM security as they increase) for these models when responding to malicious prompts with triggers are shown in Table 2.
>
> |SPPFT|LLama-3-8B-Instruct|Llama-2-7BChat|Phi-3-mini-4k-instruct|gemma-2b-it|
> |-|-|-|-|-|
> |harmful rate|8.3%|5.6%|5.2%|9.0%|
> |harmful score|1.28|1.20|1.20|1.31|
>
> *Table 2. Harmful rate and harmful score of LLMs' responses to malicious questions with instruction trigger (Tell me) after SPPFT using dataset $D_t$.*
>
>
>
> Compared to the initial aligned models, the harmful rate of the SPPFT-aligned LLMs increased by only 4.2%. This demonstrates that SPPFT effectively mitigates the security degradation caused by backdoor fine-tuning, regardless of whether the backdoor data includes triggers in the instruction. We have included the performance analysis of SPPFT on backdoor data with triggers in Appendix A.4.5 of our revised version.
>
> Thank you for your question! We hope our explanation addresses your concerns and meets your expectations.

---

> ### Author Response · Authors · 2024-11-22
> **Responses to Reviewer AP6g - Part 4/4**
>
> Responses to Reviewer AP6g - Part 4/4
>
> **[Question 1].  Are you considering an opensource model fine-tuning scenario (Rosati et al,2024) or a fine-tuning-as-a-service scenario (Qi et al, 2023)?**
>
> ...
>
> **[Question 2]. In the fine-tuning-as-a-service scenario, why you consider a backdoor attack scenario? What is the motivation for the attacker to unalign the model? There are only one motivation I can envision:**
>
> ...
>
> --------
>
>
>
> Thank you for your question!
>
> First, we want to clarify that our method can be applied to both scenarios. Below, we describe how SPPFT can be implemented in the open-source model fine-tuning scenario and the fine-tuning-as-a-service scenario:
>
> >**Open-source Model Fine-tuning Scenario**
>
> In this scenario, SPPFT can assist benign users in securely fine-tuning models. Users in the open-source LLM community typically download the weights of aligned LLMs from the community and aim to fine-tune these models locally using their own datasets. These users do not intend to compromise the model’s security alignment. However, unfortunately, even when using benign normal or backdoor datasets, alignment degradation may still occur (Qi et al., 2023).
>
> Our method provides an effective defense against such security degradation for these benign users. By following our safety layers localization pipeline to identify the safety layers, users can freeze the parameters of these layers during fine-tuning. This allows them to fine-tune the model with their own benign datasets while maintaining the alignment and security of the resulting LLM.
>
> >**Fine-tuning-as-a-Service Scenario**
>
> In this scenario, we consider the perspective of a secure fine-tuning API provider. The provider aims to offer a safe fine-tuning interface, where fine-tuning an aligned LLM with user-provided data results in minimal security degradation, ensuring that the fine-tuned model remains relatively secure. SPPFT is an excellent choice for this purpose: the API provider only needs to identify the safety layers range of the aligned LLM being fine-tuned. By applying SPPFT, they can provide users with a secure fine-tuning service.
>
> **Motivation of backdoor attacker:**  In the fine-tuning-as-a-service scenario, the motivations of backdoor attackers and harmful attackers are identical: both aim to upload specific data to fine-tune the aligned LLM into an insecure one. In the case of backdoor data, the content itself is benign, but the focus is on adding a positive response prefix to the output, such as "Sure, the answer is:".
>
> **Why we consider backdoor attack:** Both backdoor attack and harmful attack poses greater risks to the model's security in the fine-tuning stage. However, comparing to harmful data, backdoor data is more covert. In the fine-tuning-as-a-service scenario, secure fine-tuning API providers can easily use content moderation APIs (e.g., OpenAI’s Moderation API: https://platform.openai.com/docs/guides/moderation) to filter harmful data and defend against harmful attacks at the data level. However, backdoor data is benign in content and can employ trigger-free backdoor paradigms (as explained in our response in Question 3), making it more subtle and difficult for standard filters to detect. For attackers, this often makes backdoor attacks a more favorable option.
>
> We have provided detailed updates to the definitions and application goals of SPPFT in both scenarios in Appendix A.4.1 of our revised version.
>
> Meanwhile, regarding your suggestion in the last paragraph of your review: *"I suggest the authors to follow the mainstream of research and position this paper as a defense against general harmful fine-tuning attacks (on either harmful data or benign data), using backdoor attacks as a special case in the experiments. I believe this should also make this paper more impactful and valuable."*
>
> We greatly appreciate this excellent recommendation. Our supplementary experiments (referenced in our response to Weakness 1) confirm SPPFT’s potential to defend against harmful data attacks. Due to the limited time during the rebuttal period, we have temporarily included the experimental results for the harmful fine-tuning scenario in the appendix.  We will fully incorporate this suggestion into the final version of our paper, ensuring that the additional content aligns with your valuable feedback.  Thank you once again for your thoughtful input!
>
> ----
>
> Overall, we sincerely thank you for your comments for our articles and your dedication to reviewing our manuscripts!

---

> ### Comment · Reviewer_AP6g · 2024-11-22
> **The definition of "backdoor attack" is wong**
>
> Hi authors,
>
> After carefully reading the rebuttal, I find that backdoor attack defined in this paper is conceptually wrong. The authors define backdoor dataset as normal question+"Sure, the answer is" prefix, which increase the tendency of the model to output active answer. However, this attack is definitely not one of the backdoor attack. For backdoor attack, there must be a backdoor trigger presented in the testing phase to trigger the target output. The attack is more like a general concept of harmful fine-tuning attack, like implicit harmful attack defined in (Qi et al, 2023).
>
>
> I can't recommend acceptance of this paper if the key concept is wrong, as this would cause very negaitive and misleading effect to the community. However, there are still several days left before the end of author-reviewer discussion. I hope that the authors can submit a revised version of paper by eliminating all the "backdoor attack" and change to "harmful fine-tuning attack" or "fine-tuning attack" term. I will check the revision before the deadline, and will potentially increase my score. Otherwise, I will need to decrease my score as this is really a serious problem. I also welcome discussion if the authors not agree of my view.

---

> > ### Author Response · Authors · 2024-11-23
> > **Response about the definition of "backdoor attack" and the revised version**
> >
> > Thank you for your comment!
> >
> > Apologies for any potential misunderstanding. In our newly updated revised version, we have incorporated your suggestion on redefining our data paradigm. In the initial submission, we referred to the paradigm "**Instruction: {Normal instruction} Output: Sure, the answer is: {Normal output}**" as a backdoor paradigm because we considered it to be a special case of backdoor data where the instruction trigger is *None*. As such, we previously defined it in our response to Question 1 as a more covert form of backdoor data. This interpretation may have caused some disagreement.
> >
> > Furthermore, building on the additional experiments we supplemented in response to Weakness 1 and Question 1, we confirmed that SPPFT also demonstrates strong defensive performance not only in harmful fine-tuning attack scenarios but also in fine-tuning attacks using backdoor data with triggers in the instruction. Based on this, and considering your previous suggestion to frame SPPFT as a defense against general fine-tuning attack scenarios, we have updated our revised version with the following details:
> >
> > In the part on SPPFT, we first introduced four distinct fine-tuning attack scenarios in Section 4.2 ("Experiment Setting"), categorized based on the type of fine-tuning data. Meanwhile, **we redefined our previously termed backdoor attack as "Implicit Attack"**. The specific description is as follows:
> >
> > 1). Normal Data Attack: The dataset contains only harmless data.
> >
> > 2). Implicit Attack: The instruction has no trigger, but the output start with a positive response.
> >
> > 3). Backdoor Attack: Similar to an implicit attack but includes a trigger in the instruction. Both backdoor and implicit attack datasets lack malicious data.
> >
> > 4). Harmful Data Attack: A mix of normal data and malicious data.
> >
> > Additionally, we updated the title of Section 4.3 from "Freeze Safety Layers during Backdoor and Normal Data Fine-tuning" to "Freeze Safety Layers during Fine-tuning Attack". Consistent with the initial version, this section primarily introduces the effectiveness of SPPFT in handling normal data fine-tuning attacks and implicit fine-tuning attacks. Due to space constraints, detailed analyses of SPPFT's performance in harmful fine-tuning attack scenarios and fine-tuning attacks using backdoor data with triggers in the instruction have been included in Appendix A.4.4 and Appendix A.4.5, respectively. We also referenced these two scenarios at the end of Section 4.3:
> >
> > *Moreover, SPPFT is an effective defense against backdoor and harmful data fine-tuning attacks, with detailed results in Appendix A.4.5 and A.4.4. The strong performance of SPPFT across these diverse scenarios highlights its versatility in addressing general fine-tuning attacks.*
> >
> > We believe the updated title for Section 4.3 and its corresponding content better reflect SPPFT's general applicability across a wide range of fine-tuning attack scenarios, aligning more closely with your feedback and suggestions.
> >
> > Furthermore, we reviewed other parts of the main text to ensure that all descriptions of SPPFT have been updated to emphasize its role as a defense against fine-tuning attack scenarios. We also verified that there are no incorrect definitions of backdoor data in the revised text. We hope the updated revised version meets your expectations.
> >
> > Once again, we sincerely apologize for any potential misunderstandings and greatly appreciate your valuable feedback!

---

> ### Comment · Reviewer_AP6g · 2024-11-23
> **Thank you for the revision**
>
> Thank you for the revision. The new four type of attack models make sense to me. Please specify them clearly in the revision (and also, clarify which results corresponds to which attack when you describe your experiment).  I will do a thorough read before the ddl, and will post a comment if I find something need clarfication. Sounds good?

---

> > ### Author Response · Authors · 2024-11-23
> > **Revised version has been submitted**
> >
> > Thank you for your reply!
> >
> > In our revised Section 4.2, we have clearly specified the four types of attacks. Due to space constraints, we present the performance of SPPFT in the normal data attack and implicit attack scenarios in the main text. The analysis of SPPFT in the backdoor data attack scenario is included in Appendix A.4.5, while the experiments for the harmful attack scenario are detailed in Appendix A.4.4. These clarifications are explicitly addressed in both the revised Section 4.2 and Section 4.3. Following your previous suggestions and feedback, the revised version demonstrate the effectiveness of SPPFT across all four scenarios, underscoring its broad applicability as a defense method against fine-tuning attacks.
> >
> > Additionally, we thoroughly reviewed other sections of the main text to ensure that all descriptions of SPPFT emphasize its role as a defense against fine-tuning attack scenarios. We also verified that the revised text contains no incorrect definitions of backdoor data. We hope the updates in the revised version meet your expectations.
> >
> > Thank you once again, and we look forward to your feedback!

---

> ### Comment · Reviewer_AP6g · 2024-11-23
> **Could you put the harmful data attack and the normal data attack into the main text?**
>
> Hello authors,
>
> I have another read of the paper. I found that you have put normal data attack and implicit attack scenarios in the main text. However, my concern is that the implicit attack is not presented in any existing papers (not even in (Qi. et al.,2023)).  Could you please put the result of  harmful data attack and benign data attack in the main text, but leave the implicit attack and the backdoor attack in the appendix? I will have another read of the paper after your revision.

---

> > ### Author Response · Authors · 2024-11-24
> > **We have included experiments for all four defined fine-tuning attack scenarios in the main text**
> >
> > Dear Reviewer AP6g:
> >
> > Thank you for your feedback! We have submitted the latest revised version. In this version, we have included experiments for all four defined fine-tuning attack scenarios in the main text to demonstrate SPPFT’s general applicability in defending against fine-tuning attacks.
> >
> > Specifically:
> >
> > In Revised Section 4.2, after introducing the four fine-tuning attack scenarios, we added details about the construction and characteristics of the fine-tuning datasets used in these experiments, along with descriptions of the experiments in the harmful fine-tuning scenario.
> >
> > In Section 4.3, we present experimental results for the Normal, Implicit, and Backdoor attack scenarios in Table 2, and provide results for different malicious data ratios (p) under harmful attack in Table 3.
> >
> > Due to space constraints, we have moved the detailed experimental results from the original "Section 4.4: Ablation Experiment" into Revised Appendix A.4.5. The key findings of the ablation experiments are summarized in the final part of Revised Section 4.3.
> >
> > Thank you again for your feedback. We hope this revised version meets your expectations.
> >
> > Best regards

---

> ### Author Response · Authors · 2024-11-26
> **Gentle reminder of the revised version submission deadline**
>
> We sincerely thank you for your time and effort in reviewing our paper, as well as for your thoughtful feedback and appreciation of this work！ We have carefully read and addressed your comments.  As the deadline for submitting the revised version is approaching, we would be happy to discuss any additional questions or suggestions you might have to further refine our paper before submission. Building on your initial positive assessment of this work, we hope the discussions and improvements we have made will further reinforce your recommendation for acceptance.

---

> ### Comment · Reviewer_AP6g · 2024-11-26
> **Response**
>
> Hi authors,
>
> I think this paper should be accpeted. I would probably adjust my score to 7, but there is no such choice for me. For now, I will like to keep my score, but I promise that I will actively participate in the reviewer-AC discussion round.  I think this paper will has a large chance to be accpeted if you can address the concern from Reviwer MfmF.

---

> > ### Author Response · Authors · 2024-12-02
> > **Thanks for your recognition and we've addressed the concerns from Reviewer MfmF**
> >
> > Thank you for your recognition and affirmation of our work! Furthermore, as your comment, "I think this paper will have a large chance to be accepted if you can address the concern from Reviewer MfmF," we also successfully addressed Reviewer MfmF's concern during the rebuttal period, which may further enhance confidence in our paper. Once again, we sincerely appreciate your recognition of our work and the constructive discussions!

---

### Official Review · Reviewer_Bcox · 2024-10-31

**Soundness:** 3
**Presentation:** 3
**Contribution:** 3
**Rating:** 6
**Confidence:** 3

**Summary:**

The paper explores which layers in aligned large language models (LLMs) enable secure responses. The authors analyze the differences in output vectors across hidden layers, revealing the existence of specific "safety layers." Based on these findings, the authors introduce Safely Partial-Parameter Fine-Tuning (SPPFT), which selectively updates parameters outside the safety layers during fine-tuning. This approach preserves model alignment throughout the fine-tuning process without sacrificing performance.

**Strengths:**

- The authors provide an insightful perspective on analyzing LLM safety.
- The paper is clearly presented and well-structured.
- The proposed SPPFT approach is shown to be effective.

**Weaknesses:**

- The conclusion lacks rigor; differences in vectors may be explained by distributional variations.
- The assumption that certain layers, rather than individual neurons within each layer, are related to safety requires more clarification.

**Questions:**

The authors compare vector similarity to demonstrate the existence of safety-related layers, but the conclusion is not entirely rigorous. Specifically, it is unclear whether the normal and malicious queries come from the same distribution. Since malicious queries tend to focus on more sensitive tasks while normal queries are often more routine, it is likely they follow different distributions. If these distributions differ, it could also explain why cosine similarity is less impacted for N-N and M-M pairs but drops significantly for N-M pairs.

While I appreciate the authors' efforts to demonstrate the presence of layers closely related to safety, I would like to understand why they treat each layer as a whole. From my perspective, it is reasonable to consider that each layer may contain both safety-related and non-safety-related neurons. In other words, rather than one specific layer focusing mainly on safety, it could be that all layers contribute to safety, with varying numbers of neurons engaged in the process.

---

> ### Author Response · Authors · 2024-11-22
> **Responses to Reviewer Bcox - Part 1/2**
>
> Thank you very much for your insightful comments! The two weaknesses and corresponding questions you raised align directly with one another. Below, we provide detailed responses to each pair of weaknesses and questions.
>
>  **[Weakness 1]. The conclusion lacks rigor; differences in vectors may be explained by distributional variations.**
>
>  **[Question 1]. The authors compare vector similarity to demonstrate the existence of safety-related layers, but the conclusion is not entirely rigorous. Specifically, it is unclear whether the normal and malicious queries come from the same distribution. Since malicious queries tend to focus on more sensitive tasks while normal queries are often more routine, it is likely they follow different distributions. If these distributions differ, it could also explain why cosine similarity is less impacted for N-N and M-M pairs but drops significantly for N-M pairs.**
>
>  ---------
>
> Thank you for your comments and thoughtful insights!
>
> Firstly, there exists a phenomenon in Figure 2 of the main text: for each aligned LLM, the average cosine similarity curves of N-N pairs and N-M pairs are nearly identical in the initial layers, with a noticeable gap only emerging from a certain intermediate layer. If the difference in processing N-N pairs and N-M pairs by aligned LLMs fundamentally stems from differences in the vector distributions of these question categories, we would expect to see a more pronounced divergence in the curves starting from the earliest layers. However, as shown in the figure, the gap between the curves only begins to widen from a specific intermediate layer, exhibiting an increasing growth rate before eventually leveling off. Thus, we interpret this phenomenon as follows: **In the initial layers, aligned LLMs process malicious and normal questions similarly. It is only in the specific intermediate layers—enabled by the effect of safety layers—that the LLM starts to distinguish between the two types of questions**.
>
> Furthermore, if the curve gap observed in aligned LLMs were solely due to differences in vector distributions, we would expect to see a similar gap in pre-trained LLMs. However, Figure 3 in the main text illustrates the N-N Pair and N-M Pair analysis for pre-trained LLMs such as LLaMA-2, LLaMA-3, and Gemma, which lack security alignment. We found that, for these pre-trained LLMs, the average cosine similarity curves for N-N Pairs and N-M Pairs remain nearly identical across all layers. This suggests that the curve gap may not stem from differences in vector distribution, but rather provides a clearer visualization of how the LLM distinguishes between different types of questions(malicious and normal). The lack of a gap in Figure 3 for pre-trained LLMs aligns with their inability to distinguish between malicious and normal questions, further indicating that the emergence of safety layers is a product of security alignment.
>
> In the revised version, we have included a discussion on this topic in Appendix A.5.4. We hope our response can address your concerns. Thank you once again for your valuable comments!

---

> ### Author Response · Authors · 2024-11-22
> **Responses to Reviewer Bcox - Part 2/2**
>
> **[Weakness 2]. The assumption that certain layers, rather than individual neurons within each layer, are related to safety requires more clarification.**
>
> **[Question 2]. While I appreciate the authors' efforts to demonstrate the presence of layers closely related to safety, I would like to understand why they treat each layer as a whole. From my perspective, it is reasonable to consider that each layer may contain both safety-related and non-safety-related neurons. In other words, rather than one specific layer focusing mainly on safety, it could be that all layers contribute to safety, with varying numbers of neurons engaged in the process.**
>
> ---------
>
> Thank you for your insightful comment! This provides a new perspective for consideration. Let me first explain why we chose to analyze the aligned LLMs layer by layer:
>
> Firstly, in our main text, we cited a study by [1], which interprets security from the neuron level. In this study, the authors identified "security neurons" by analyzing neuron responses to malicious prompts and found that removing only the top 3% of these neurons significantly compromises model security. However, they also showed that removing these neurons substantially degrades the LLM's overall performance, and freezing security neurons alone does not defend against fine-tuning attacks.  Motivated by these insights, we sought to explore a new perspective for analyzing LLM security beyond the neuron level.
>
> Secondly, in our layer-wise analysis of the cosine similarity curve between N-N pairs and N-M pairs for aligned LLMs, layer-based analysis revealed a clear distinction. Additionally, in Section 3.6 of the main text, our layer-by-layer analysis of attention scores revealed an interesting hierarchical effect: layers before the safety layers do not focus on tokens strongly related to semantics, whereas only layers after the safety layers begin to focus on task-related instruction tokens. This provides a meaningful layer-based perspective on LLM utility. (More attention score heatmaps can be found in Appendix A.5.2.)
>
> Last but not least, in the experimental section of the main text, we demonstrated the effectiveness of freezing parameters of the safety layers (SPPFT) for preserving the security of aligned LLMs. Furthermore, in the ablation study (Section 4.4), we experimented with freezing other contiguous layers outside the safety layers and found that only SPPFT was effective. The comparison between SPPFT and ablation experiment further explains the rationality of layer-wise analysis.
>
> In summary, analyzing security at the layer level provides a feasible and insightful approach to understanding LLM security.
>
> Furthermore, I believe your perspective that "rather than one specific layer focusing mainly on safety, it could be that all layers contribute to safety, with varying numbers of neurons engaged in the process" is also valid, though it differs from our analytical aspect. Following your viewpoint, our experimental findings may suggest that the neurons responsible for security in aligned LLMs are more concentrated in a specific subset of contiguous layers. Freezing these layers during fine-tuning appears to better preserve the functionality of these neurons, thereby defending against the degradation of security alignment during fine-tuning. This is an insightful perspective, and we have included the above discussion in Appendix A.5.4 of of our revised version.
>
> [1] Boyi Wei, et al.  "Assessing the Brittleness of Safety Alignment via Pruning and Low-Rank Modifications", Proceedings of the 41 st International Conference on Machine Learning
>
> ----
>
> Finally, thank you sincerely for your valuable comments on our article and for your time and dedication in reviewing our manuscript. We hope that, building on your initial positive evaluation, the discussions we've made in response to your feedback will further convince you of the paper's merit. Thank you once again!

---

> > ### Comment · Reviewer_Bcox · 2024-11-26
> > **Thank you for the response**
> >
> > Thank you for the response. I agree that studying simple neurons may not be a good option as it overlooks the synergy. However, extending this idea a bit, it is still possible that a subnetwork, like a "lottery ticket," exists and plays a vital role in security. As I am already positive about this paper, I will maintain my score.

---

### Official Review · Reviewer_oVJL · 2024-11-04

**Soundness:** 3
**Presentation:** 2
**Contribution:** 3
**Rating:** 6
**Confidence:** 3

**Summary:**

This paper identifies specific "safety layers" within language models that play a critical role in differentiating between malicious and normal queries. These safety layers display noticeable distributional discrepancies between benign and adversarial queries, providing an insight into potential defense mechanisms. Leveraging this finding, the authors propose a fine-tuning approach called Safely Partial-Parameter Fine-Tuning (SPPT) to enhance model resilience against certain threats.

**Strengths:**

- The identification of a specific set of middle layers in aligned LLMs as key to recognizing malicious inputs is an intriguing finding that provides a new perspective on model robustness.
- The paper presents clear and effective visualizations to support the findings.
- Experiments demonstrate the method’s efficacy.

**Weaknesses:**

- The type of attacks addressed by the proposed method is unclear. While related work discusses jailbreak, the experiments primarily use backdoor datasets. The paper lacks a defined threat model and discussion of the defender's capabilities and objectives.
- Since the method requires access to model parameters, it is not suitable for popular black-box LLMs, which limits its applicability.

**Questions:**

Please see weakness

---

> ### Author Response · Authors · 2024-11-22
> **Responses to Reviewer oVJL - Part 1/2**
>
> Many thanks for the insightful and helpful comments by the reviewer! We make the following point-by-point responses to your comments:
>
> **[Weakness 1]. The type of attacks addressed by the proposed method is unclear. While related work discusses jailbreak, the experiments primarily use backdoor datasets. The paper lacks a defined threat model and discussion of the defender's capabilities and objectives.**
>
> -------
> [Answer 1] Part 1/2:
>
> Thank you for your insightful comments! To clarify the scenario we are addressing, we first define the threat model along with the defender's capabilities and objectives. Additionally, we supplemented the SPPFT experiments in the harmful fine-tuning jailbreak scenario.  The details are as follows:
>
> **---Threat Model---**
>
> **1. Attacker Identity and Goals**
>
> **Identity:** The attacker is a user of the fine-tuning API.
>
> **Goal:** By providing one specially crafted fine-tuning dataset, the attacker aims to get a LLM with compromised security.
>
> **2. Attacker Capabilities**
>
> **Data Upload Permissions:** The attacker can upload any dataset to the fine-tuning API but cannot bypass the API’s content moderation mechanisms.
>
> **Data Manipulation:** The attacker is capable of carefully designing datasets to influence the behavior of the fine-tuned model, including using seemingly benign but subtly crafted backdoor data.
>
> **Limited Knowledge:** The attacker lacks knowledge of the model’s internal structure, parameters, and the location of safety layers and cannot directly access or modify model parameters.
>
> **Restricted by Filtering Mechanisms:** The attacker cannot upload explicitly malicious content, as such content would be flagged and filtered by the defender’s content moderation API.
>
> **---Defender's Capabilities and Objectives---**
>
> **1. Defender Identity and Goals**
>
> **Identity:** The defender is the provider of the fine-tuning API, responsible for maintaining the model's security and performance.
>
> **Goal:** Ensure that, regardless of the type of data provided by users, the fine-tuned model maintains a high level of security and prevents the generation of harmful content.
>
> **2. Defender Capabilities**
>
> **Model Access:** The defender has access to the parameters and the structure of the LLM, including the location and functionality of the safety layers.
>
> **Malicious Content Moderation:** The defender can use content moderation APIs (e.g., OpenAI’s Moderation API: https://platform.openai.com/docs/guides/moderation) to filter out malicious or harmful user-uploaded data.
>
> **Fine-Tuning Control:** The defender can implement secure fine-tuning strategies (such as SPPFT), freezing the parameters of the safety layers during fine-tuning to prevent a decline in security performance.
>
> Based on the scenario described above, our experiments in the main text primarily focus on the security of fine-tuned LLMs when attackers use either backdoor data or normal data. The results demonstrate that, compared to full fine-tuning, SPPFT significantly preserves model security while maintaining its performance.
>
> We have provided detailed updates to the definitions and application goals of SPPFT in both scenarios in Appendix A.4.1 of our revised version.

---

> ### Author Response · Authors · 2024-11-22
> **Responses to Reviewer oVJL - Part 2/2**
>
> [Answer 1] Part 2/2:
>
> **Harmful data fine-tuning jailbreak:**
>
> Regarding your comment that "the experiments primarily use backdoor datasets", we further assessed SPPFT’s performance in a broader malicious fine-tuning jailbreak scenario, complementing the evaluation of its defensive effectiveness on backdoor and normal datasets as presented in the main text. Specifically, we evaluated SPPFT’s performance when the fine-tuning dataset consisted of harmful data mixed with normal data. Specific experimental settings are as follows:
>
> The fine-tuning dataset consisted of 1,000 normal data samples and 1,000 * p malicious data samples, where p represents the percentage of malicious data. The initial learning rate for fine-tuning was set to 1e-5, and all other experimental parameters were consistent with those in our original submission.
>
> We evaluated the harmful rate and harmful score of models fine-tuned using FullFT and SPPFT under the conditions *p* = 0.05, *p* = 0.1, and *p* = 0.2. Higher values of these two metrics indicate a lower level of model security. We use LLaMA-2-chat and  Phi-3-mini-4k-instruct as the fine-tuning objects respectively, and the detailed experimental results are summarized in Table 1.
>
> | | |FullFT| |SPPFT| |
> |-|-|-|-|-|-|
> | | |**Harmful Rate**|**Harmful Score**|**Harmful Rate**|**Harmful Score**|
> |**LLaMA-2-chat**|p=0.05|26.0%|1.88|7.9%|1.25|
> | |p=0.1|53.7%|2.97|25.6%|1.86|
> | |p=0.2|93.5%|4.53|68.3%|3.61|
> |**Phi-3-mini-4k-instruct**|p=0.05|59.4%|3.23|8.1%|1.29|
> | |p=0.1|90.7%|4.45|23.3%|1.74|
> | |p=0.2|98.8%|4.89|78.8%|3.88|
>
> *Table 1. Under the Settings of p=0.05, p=0.1 and p=0.2, the harmful rate and harmful score of the LLMs obtained after FullFT and SPPFT were adopted, respectively.*
>
> The experimental results indicate that SPPFT can significantly mitigate the security degradation caused by harmful data fine-tuning compared to FullFT under different values of malicious data rate (p).  This confirms SPPFT’s potential to protect model security in general harmful fine-tuning attack scenarios.
>
> Thank you again for your valuable comments! We have included the experimental performance of SPPFT in the harmful fine-tuning jailbreak scenario in Section 4.3 of our revised version.
>
>
>
>
> **[Weakness 2]. Since the method requires access to model parameters, it is not suitable for popular black-box LLMs, which limits its applicability.**
>
> -------
>
> Thank you for your comments on the application scenarios of SPPFT!
>
> Firstly, from the defender's perspective (in this case, the fine-tuning API provider), our proposed SPPFT method serves as a white-box defense. Its goal is to protect against black-box attacks on large language models (LLMs).  The detailed threat model as well as the defender's capabilities and objectives is presented in our response to Weakness 1.
>
> Additionally, in open-source model fine-tuning scenario, SPPFT can assist benign users in securely fine-tuning models. Users in the open-source LLM community typically download the weights of aligned LLMs from the community and aim to fine-tune and deploy these models locally using their own datasets. These users do not intend to compromise the model’s security alignment. However, unfortunately, even when using benign normal or backdoor datasets, alignment degradation may still occur ([1] Qi et al., 2023). Our method provides an effective defense against such security degradation for these benign users. Since both fine-tuning API providers and open-source benign users have access to the model's parameters, SPPFT has a wide range of applicability and value in this white-box scenario.
>
> Last but not least, our experimental results in the main text demonstrate its strong defensive effectiveness in backdoor and normal fine-tuning scenarios under white-box conditions. Furthermore, our analysis in the response to Weakness 1 expands SPPFT’s applicability to harmful data fine-tuning scenarios, further broadening its range of use.
>
> Overall, SPPFT is a simple yet effective method for defending against fine-tuning attacks, with broad applicability in white-box scenarios.
>
> [1]. Xiangyu Qi et al, "Fine-tuning Aligned Language Models Compromises Safety, Even When Users Do Not Intend To!", ICLR2024
>
> ----
>
> We sincerely appreciate your valuable feedback and the time you dedicated to reviewing our manuscript. Building on your initial positive assessment of this work, we hope that the discussions and improvements we have made in response to your comments will further strengthen your recommendation for acceptance. Thank you once again for your thoughtful and constructive input!

---

### Public Comment · ~Yige_Li1 · 2024-11-13
**Missing Citation for the "Safety Layer" Concept**

Thanks for this nice work. As a friendly reminder, the concept of the "safety layer" [1] was first introduced in our paper published at EMNLP 2024. Given the similarity in the focus on the "safety layer" between our works, we hope the authors could consider citing our study. Thank you again!

[1] Defending Large Language Models Against Jailbreak Attacks via Layer-specific Editing, EMNLP 2024

---

> ### Author Response · Authors · 2024-11-20
> **Response to public comment**
>
> Thank you very much! We will cite your work in our final version as a related concurrent work that uses the same term "safety layer" for different purposes. Additionally, we will discuss the differences between the two approaches, including the different definitions of safety layers, how to identify them, and their applications. We appreciate your suggestions regarding this related work. Thanks again!

---

### Meta-Review · Area_Chair_Doxt · 2024-12-21

**Metareview:**

This paper study the ``safety layers'' in models. Motivated by the safety layers, it further proposes a fine-tuning methods which fix the gradient of safety layers during fine-tuning to improve the safety. Before rebuttal, some reviewers have concerns about the definition of backdoor attacks, presentation, experimental results. After rebuttal, the authors did a good job to address the concerns. All reviewers can provide positive scores and one reviewer also gave score 7 (although the reviewer can not update the score). AC read all reviewers comments and agreed with their scores. AC hopes authors could add them in the revised version.

**Additional Comments On Reviewer Discussion:**

The main concerns from reviewer oVjL are (1) the type of attack is unclear, no threat model, (2) method requires to access the model, which is not suitable for blackbox LLM. AC read the authors rebuttal and the authors addressed the comments well.

Concerns from BcoX are (1) conclusion lacks rigor and (2) assumption requires more clarifications. After rebuttal, reviewer agreed that studying simple neurons may not be a good option as it overlooks the synergy and provided positive feedbacks.

Concerns from Reviewer AP6g are (1)  missing experiments, (2)baselines, (3) literature review, (4) unclear experimental setting. During the rebuttal, reviewer also pointed out the wrong definition about backdoor attacks. Finally, reviewers have addressed all concerns. AC read the rebuttal and agreed with reviewer's positive feedback.

Concerns from Reviewer MfmF are (1) experimental setting; (2) evaluation metric; (3) missing reference; (4) unclear explanation about experimental results. After the rebuttal, reviewer provides the positive score.

---

### Decision · Program_Chairs · 2025-01-22

Accept (Poster)